# *Cissus quadrangularis* (Hadjod) Inhibits RANKL-Induced Osteoclastogenesis and Augments Bone Health in an Estrogen-Deficient Preclinical Model of Osteoporosis Via Modulating the Host Osteoimmune System

**DOI:** 10.3390/cells12020216

**Published:** 2023-01-04

**Authors:** Zaffar Azam, Leena Sapra, Kalpana Baghel, Niharika Sinha, Rajesh K. Gupta, Vandana Soni, Chaman Saini, Pradyumna K. Mishra, Rupesh K. Srivastava

**Affiliations:** 1Translational Immunology, Osteoimmunology & Immunoporosis Lab (TIOIL), Department of Biotechnology, All India Institute of Medical Sciences (AIIMS), New Delhi 110029, India; 2Department of Zoology, Dr. Harisingh Gour Vishwavidyalaya (Central University), Sagar 470003, India; 3Drug Development Laboratory, School of Vocational Studies and Applied Sciences, Gautam Buddha University, Greater Noida 201312, India; 4Department of Pharmaceutical Sciences, Dr. Harisingh Gour Vishwavidyalaya, Sagar 470003, India; 5Department of Molecular Biology, ICMR-NIREH, Bhopal 462001, India

**Keywords:** *Cissus quadrangularis*, osteoporosis, immunoporosis, Th17, Tregs, Bregs

## Abstract

Osteoporosis is a systemic skeletal disease characterised by low bone mineral density (BMD), degeneration of bone micro-architecture, and impaired bone strength. *Cissus quadrangularis* (CQ), popularly known as Hadjod (bone setter) in Hindi, is a traditional medicinal herb exhibiting osteoprotective potential in various bone diseases, especially osteoporosis and fractures. However, the cellular mechanisms underpinning its direct effect on bone health through altering the host immune system have never been elucidated. In the present study, we interrogated the osteoprotective and immunoporotic (the osteoprotective potential of CQ via modulating the host immune system) potential of CQ in preventing inflammatory bone loss under oestrogen-deficient conditions. The current study outlines the CQ’s osteoprotective potential under both ex vivo and in vivo (ovariectomized) conditions. Our ex vivo data demonstrated that, in a dose-dependent manner CQ, suppresses the RANKL-induced osteoclastogenesis (*p* < 0.001) as well as inhibiting the osteoclast functional activity (*p* < 0.001) in mouse bone marrow cells (BMCs). Our in vivo µ-CT and flow cytometry data further showed that CQ administration improves bone health and preserves bone micro-architecture by markedly raising the proportion of anti-osteoclastogenic immune cells, such as Th1 (*p* < 0.05), Th2 (*p* < 0.05), Tregs (*p* < 0.05), and Bregs (*p* < 0.01), while concurrently lowering the osteoclastogenic Th17 cells in bone marrow, mesenteric lymph nodes, Peyer’s patches, and spleen in comparison to the control group. Serum cytokine analysis further supported the osteoprotective and immunoporotic potential of CQ, showing a significant increase in the levels of anti-osteoclastogenic cytokines (*p* < 0.05) (IFN-γ, IL-4, and IL-10) and a concurrent decrease in the levels of osteoclastogenic cytokines (*p* < 0.05) (TNF-α, IL-6, and IL-17). In conclusion, our data for the first time delineates the novel cellular and immunological mechanism of the osteoprotective potential of CQ under postmenopausal osteoporotic conditions.

## 1. Introduction

Osteoporosis is a progressive, systemic skeletal disease characterised by low BMD, degeneration of bone micro-architecture, and impaired bone strength which makes the bone prone to developing fragility-related fractures. Among the elderly population, osteoporosis is particularly prevalent in postmenopausal women and patients undergoing long-term steroidal medication. Due to the lack of clinical symptoms before the development of fractures, the disease is frequently referred to as “silent”. Recent data available from the International Osteoporosis Foundation (IOF) suggest that over 200 million individuals are presently living with complications of osteoporosis. Furthermore, according to epidemiological data, one in three women and one in five men over the age of 50 will sustain an osteoporotic-related fracture at least once in their lives [1] thus contributing to a huge economic burden on the health care systems. Numerous variables, including a diet low in calcium and vitamin D, smoking, alcoholism, and a lack of oestrogen and parathyroid hormone, contribute to the development of a variety of bone illnesses, including osteoporosis, osteoarthritis, osteomyelitis, and Paget’s disease [2,3]. To treat osteoporosis and other related bone conditions, a variety of anti-osteoclastogenic and bone-building drugs are widely employed [4]. In addition, hormone replacement therapy (HRT) with estrogen or progesterone is an alternative proven method for the prevention of postmenopausal osteoporosis (PMO). However, taking any of the above-mentioned drugs, including HRT, can have a number of unfavorable side effects, including depression, osteonecrosis of the jaw, hypercalciuria, breast discomfort, an elevated risk of endometrial and breast cancer, and cardiac stroke. Moreover, most of these medications have been reported to enhance the risk of bone fragility-related fractures over the long run [5,6,7]. Thus, novel, efficient, and economical therapies with few or no side effects are needed.

Research from our group along with others in the field of osteoimmunology had already proven that immune cells including Th1, Th2, Tregs, Th17, and Bregs play a crucial role in the pathogenesis of several inflammatory and autoimmune bone-related diseases, including osteoporosis [8,9,10,11,12,13]. Due to their anti-osteoclastogenic capabilities, Th1 and Th2 cells promote bone formation by secreting anti-osteoclastogenic cytokines like IFN-γ and interleukin (IL)-4, respectively. Recently, our team has demonstrated how vital the homeostatic balance between Tregs and Th17 cells is for maintaining bone health under osteoporotic circumstances [14,15,16]. Of note, Th17 cells enhance osteoclastogenesis via secreting the proinflammatory (IL-6, IL-17, and TNF-α) cytokines. Conversely, Treg cells limit bone resorption by secreting anti-osteoclastogenic cytokines (IFN-γ, IL-4, and IL-10) [14,15,17,18,19,20]. Of late, our group established the inhibitory role of Bregs on osteoclastogenesis along with ameliorating inflammatory bone loss under postmenopausal conditions in an IL-10-mediated manner [21]. Counterintuitively, it has also been shown that Bregs control the differentiation of Tregs and Th17 cells. According to a recent study, the adoptive transfer of Bregs reduces the proportion of IL-17 producing Th17 cells, hence preventing alveolar bone loss. Altogether, these studies including our own reflect the pivotal role of the “Breg-Treg-Th17” cell axis in regulating bone remodeling, and pharmacological exploitation of this axis could be employed for managing and treating inflammatory bone loss under oestrogen-deficient conditions [22]. These investigations establish that immune cells play a key role in the pathogenesis of osteoporosis, thereby paving the path for a unique field of biology proposed and named by our group as “Immunoporosis” (i.e., Immunology of Osteoporosis) [19].

The present understanding of several plants and their bioactive components (phytoconstituents) with beneficial medicinal and therapeutic qualities suggests their significant involvement in the management and treatment of various inflammatory bone diseases, such as osteoporosis [23]. CQ is a succulent climber of the family Vitaceae generally characterized by its angular stems. It is mainly found in America, Australia, India, Southeast Asia, and Africa [24]. Scientific interest in this species is increasing rapidly because of its valuable contribution both to veterinary and human medicine [25]. Owing to its medicinal properties, it has been widely used since ancient times in both Ayurvedic and Siddha systems of medicines as an analgesic, general tonic, and bone fracture healing [26].

Oral administration of a variety of CQ extracts [27,28,29] and its phytoestrogenic steroids [30,31] augments bone health via early regeneration and quick mineralization in an animal model of osteoporosis, highlighting the beneficial effects of CQ on bone health. However, no study has yet identified and demonstrated the osteoimmunological mechanism by which CQ inhibits inflammatory bone loss in estrogen-deficient settings. Based on this evidence, we examined the osteoprotective and immunoporotic potential of CQ. Altogether, results from the present study provide compelling evidence in favor of the anti-osteoclastogenic and immunoporotic (the osteoprotective potential of CQ via modulating the host immune system) potential of CQ in ameliorating inflammatory bone loss in a preclinical model of PMO. 

## 2. Materials and Methods

### 2.1. Animals and Treatments

For our in vivo experiments, female Balb/c mice, age (8–10 weeks), and weight (23 ± 2 gm) were procured and housed in polystyrene cages at the institutional animal facility (Department of Pharmaceutical Sciences, Dr. Harisingh Gour Vishwavidyalaya, Sagar, Madhya Pradesh, India), under a specific pathogen-free (SPF) atmosphere with constant humidity (45–65%), temperature, and a 12-h light-dark cycle. Experimental mice had unlimited access to a conventional mouse diet and water. Mice were acclimatized for one week in laboratory conditions before ovariectomy. Post-acclimatization mice underwent ventral bilateral ovariectomy (Ovx), after anesthetizing intraperitoneally with Ketamine (100–150 mg/kg) and Xylazine (5–16 mg/kg) according to the standardized protocols and procedures. One-week post-surgery, mice were randomly divided into three groups, i.e., Sham-operated, Ovx, and Ovx + CQ with six mice/group. 

The numbers of animals used in our experiments were based on statistical power calculations with parameters from our extensive previous bone health-related experiments in our lab. Bone Mineral Density (BMD) was used as the parameter for power calculations. From our previous study (Dar et al., 2018), we found that the BMD of the bone of ovx is 1.74 gm HA/cm^3^ (mean) and the BMD of the bone of sham mice is 2.8 gm HA/cm^3^ (mean). Standard Deviation (SD) for this experiment was 0.65 gm HA/cm^3^. From this data, we selected the level of significance at 5% and the power of study at 80%. The effect size (mean of two groups) in this situation is 1.03. Hence, the sample size was calculated as follows: sample size = 2SD^2^ (Z score value of level of significance (5%) + (Z score value of power of the study (80%) / effect size = 2 x 0.4225 (1.96 +0.842)^2^/(1.03)^2^ = 6.44. Thus, we arrived at using six mice per group in our current study. 

The Ovx + CQ group mice were orally administered with *Cissus quadrangularis* crude extract at a dose of 500 mg/kg/day using mice oral feeding needles, twice daily for 45 days. On the final day of the experiment, mice were euthanized via CO_2_ asphyxiation, followed by the collection of bone, lymphoid tissues, and blood samples through retro-orbital sinus for further analysis. All the investigational procedures were followed according to the prior permission available with the submitted protocols to The Committee for Control and Supervision of Experiments on Animals (CPCSEA), India with former sanction from Institutional Animal Ethical Committee (IAEC), Dr. Harisingh Gour Vishwavidyalaya, Sagar, Madhya Pradesh, India, with CPCSEA, Registration No. (379/CPCSEA/IAEC-2018/017).

### 2.2. Collection of Plant Material

Excellent quality CQ stems were collected from the Botanical Garden, Dr. Harisingh Gour Vishwavidyalaya, (A Central University), Sagar, (M.P.), India, and authenticated at the Department of Botany, Dr. Harisingh Gour Vishwavidyalaya, Sagar, (M.P.), India. The voucher specimen no. BOT/H/02/50/04 has been kept in the herbarium file of the same department.

### 2.3. Preparation of CQ Extract

CQ stems weighing 1.5 kg were collected, washed (tap water), air-dried, and ground into a coarse powder. The powdered stems were then extracted with 90% (v/v) ethanol by using the Soxhlet apparatus at 60 °C [32,33]. The resultant CQ crude extract was filtered, and the extra solvent was removed by a vacuum evaporator for experiments. The yield was calculated (150 g) with a syrup-like consistency and the extract was stored at 4 °C for further investigations. 

### 2.4. Reagents and Antibodies 

The mentioned antibodies and reagents such as PerCp-Cy5.5 Anti-Mouse-CD4-(RM4-5) (550954), APC Anti-Mouse/Rat-Foxp3 (FJK-16s) (17-5773), PE Anti-Human/Mouse-Rorγt (AFKJS-9) (12-6988), PerCp-Cy5.5 Anti-Mouse-CD19 (1D3) (45-0193-82), PE-Cy7 Anti-Mouse-CD5 (53-7.3) (25-0051-81), APC Anti-Mouse-CD1d (1B1) (17-0011-82), BV605 Anti-Mouse-IFN-γ (XMG1.2) (505839), BV711 Anti-Mouse-IL-4 (11B11) (504133), Foxp3/Transcription factor staining buffer (0-5523-00), and RBC lysis buffer (00-4300-54) were purchased from eBiosciences San Diego, California, (USA). The subsequent ELISA kits i.e., Mouse IL-17 (M1700) and Mouse IL-10 (M1000B) Quantikine ELISA kits were procured from R & D, Minneapolis (USA). The following ELISA kits and reagents, i.e., Mouse TNF-α (OptEIA-560478) and Mouse IL-6 (OptEIA-555240) were procured from BD, Franklin Lakes, New Jersey (USA). Acid phosphatase, leukocyte (TRAP) kit (387A), 4′6-diamidino-2-phenylindole (DAPI), (D7592), and FITC-Phalloidin (P5282) were purchased from Sigma Aldrich (USA). Receptor activator of nuclear factor κβ-ligand (sRANKL) (CYT-334) and macrophage colony-stimulating factor (MCSF) (CYT-308) were procured from ProSpec (Israel). Alpha minimal essential media (α-MEM) was purchased from Gibco, Waltham, MA, (USA).

### 2.5. Osteoclasts Differentiation and TRAP Staining 

Primary bone marrow-derived macrophages (BMMs) were obtained from the mouse bone marrow cavity by flushing both tibia and femur dissected out of the 8-10-week-old, female Balb/c mice, as reported previously [16,34]. Briefly, the bone marrow cells were further suspended in complete α-MEM media containing 10% heat-inactivated fetal bovine serum (FBS). Subsequently, red blood cells (RBCs) were lysed by using 1X RBC lysis buffer, followed by overnight culturing of cells in complete α-MEM media supplemented with MCSF (35 ng/mL). The non-adherent cells were collected and the following day and seeded in an osteoclastogenic medium with RANKL (100 ng/mL) and MCSF (30 ng/mL) with or without CQ extracts at different concentrations, i.e., 10, 20, 50, and 100 g/mL for f consecutive days. On the third day, the culture media was replenished by replacing half of the media with media containing fresh osteoclastogenic factors. To determine the presence of mature multinucleated osteoclasts (with >03 nuclei), cells were then treated with a fixative solution containing formaldehyde, acetone, and citrate for 10 min at 37 °C. Following fixation, cells were stained with TRAP according to the manufacturer’s instructions. Osteoclasts with a nucleus count of ≥ 3 were regarded as mature osteoclasts. An inverted microscope (ECLIPSE, TS100, Nikon, Japan) was used to count and capture images of multinucleated TRAP^+^ cells. Image-J (NIH, Bethesda, MD, USA) software was used to count and measure the area of TRAP^+^ cells. 

### 2.6. F-Actin Ring Polymerization Assay

The F-actin ring polymerization experiment was performed as previously described [34]. Specifically, bone marrow-derived monocytes/macrophages (osteoclast precursors) were seeded on glass coverslips in 12-well plates with or without CQ at two distinct concentrations, i.e., 50 and 100 μg/mL. On day 4, cells were rinsed twice with 1X PBS and fixed with 4% paraformaldehyde (PFA) for 20 min. The cells were then treated with 3% BSA for 30 min to prevent non-specific binding, and then stained with FITC-labelled-phalloidin for an hour at room temperature in the dark. Finally, cells were stained for 5 min in the dark with DAPI (10 μg/mL), and the slides were further examined and imaged using an immunofluorescence microscope (Imager Z2, Zeiss).

### 2.7. Cell Viability or Metabolic Activity Assay

To determine the viability of the cells in relation to their metabolic activity, the MTT test was used. Briefly, BMMs were seeded in 96-well plates at a density of 10,000 cells per well and cultured for 24 h at 37 °C in a CO_2_ incubator. Cells were exposed to CQ at various doses the next day and were further incubated for 48 h. Following the completion of the treatment, MTT (0.5 mg/mL) was added, and the plate was incubated in a CO_2_ incubator at 37 °C for 4 h. After incubation, DMSO was used to dissolve the formazan crystals. The plate was orbitally shaken for 5 s, and a reading at 570 nm was obtained using a microplate reader (Synergy H1, BioTek)

### 2.8. Scanning Electron Microscopy (SEM)

SEM images of the cortical regions of the femur bones from distinct groups were collected in accordance with the previous description [14,15]. Briefly, the femur cortical bones dissected were submerged in triton X-100 (1%), for 2–3 days. After the completion of 3 days, the bone samples were again resuspended in 1X PBS for further investigations. Bone slices were cut and stored under an incandescent lamp for one day before to analysis to dry the bone samples. The dried bone samples were then sputter-coated with platinum. Next, the platinum-coated samples were analyzed by a Leo 435-VP SEM outfitted with a 35 mm camera and digital imaging. The final step was to collect SEM images of the targeted cortical regions from the comparable regions of the various groups at three different magnifications i.e., 10X (lower) 1000X (intermediate), and 10,000X (higher). MATLAB software was further used to analyze the captured images (Math Works, Natick, MA, USA). 

### 2.9. Micro-Computed Tomography (µ-CT) 

To analyze the trabecular areas of the femur, tibia, and lumbar vertebrae-5 (LV-5) as previously described, μ-CT was carried out using a Sky-Scan 1076 scanner (Aartselaar, Belgium) [14,34]. After loading the samples, a 0.5 mm aluminum filter was applied, the exposure time was adjusted to 590 ms, and scanning was done at 50 Kv, 201 mA, with each sample oriented precisely for analysis. With a resolution of 9 m/pixel, 1800 pictures in total were collected. The reconstruction of the desired bone samples was then finished using NRecon software. Next, for performing the segregation of growth plates from the adjoining tissues, manual segmentation was carried out to obtain 2D segments of sagittal radiographs. It was then preceded by the reconstruction step for gaining the 3D µ-CT images. Next, the images obtained were analyzed for choosing the growth plate’s true height with the help of the data viewer tool. Region of interest (ROI) was drawn in secondary spongiosa for a total of 100 slices to analyze the trabecular bones. The primary spongiosa and cortical portions of the bone samples were excluded, and it was positioned 1.5 mm from the distal end of the growth plates. Similar to this, a total of 50 uninterrupted slices were included for the in vivo measurements of the LV-5 trabeculae, starting from the beginning of the trabecular region of the bone within the vertebral body [35,36]. To further examine the cortical bone micro-architectural details, 350 subsequent image slices were discarded from the trabecular area of the bone behind the growth plates. Finally, only 200 consecutive image slices out of these 350 slices were chosen for further analysis utilizing the CTAn software in vivo. Various 3D-histomorphometric trabecular analyses such as BV/TV (bone volume/tissue volume), Tb.Sp (trabecular separation), Tb.Th (trabecular thickness), and 2D cortical parameters like Tt.Ar (total cross-sectional area inside the periosteal envelope), Ct.Ar/Tt.Ar (cortical area fraction), and Ct.Th (cortical thickness) was considered [18,35,37]. The BMD of the femur, tibia, and LV-5 regions of the bones were then estimated using the volume of interest (VOI) gathered for μ-CT scans made from both cortical and trabecular portions of the bones. The results were obtained by calibrating using hydroxyapatite phantom rods with a 4 mm diameter and known BMD values: 0.25/cm^3^ and 0.75/cm^3^ [14,15]. As a result, the estimated BMD was determined for each examination of the desired bone samples using the linear relationship between BMD and µ-CT attenuation coefficient.

### 2.10. Flow Cytometry

Immediately following mice sacrifice, the lymphoid tissues such as BM, SP, MLN, and PP were harvested and homogenized to make a single cell solution. Then, RBCs were lysed with 1X RBC lysis buffer and washed twice with 1X PBS. Further cells were processed for immune cell staining. For analysis of intracellular cytokines, we followed the following protocol: cells obtained from different lymphoid organs were resuspended in RPMI-1640 complete medium containing 10% FBS and seeded in 96 well plates at a density of 1 × 10^6^ cells/mL. Cells were further stimulated with a mixture of Ionomycin (1 ug/mL), and PMA (50 ng/mL, Sigma Aldrich). Before harvesting, cells were treated with protein transport inhibitors (BD, USA) containing brefeldin and monensin for 5 h. After harvesting, stimulated cells were stained for both surface and intracellular markers.

Flow cytometry analysis was carried out by the previously defined and stated methodologies [16,21]. The flow cytometry was performed for evaluating the percentage of Bregs, Tregs, Th17, Th1, and Th2 cells [18,37]. First, cells were surface stained for Bregs (anti-CD19-PerCP-Cy5.5, anti-CD5-PE-Cy7, and anti-CD1d-APC antibodies) and Tregs/Th17/Th1/Th2 (anti-CD4-PerCP-Cy5.5) in two separate tubes and incubated for 30 min in the dark on ice. After surface staining, cells were washed, fixed, and permeabilized with 1X fixation permeabilization buffer. Finally, intracellular staining was performed, and cells were further stained with APC-conjugated-anti-Foxp3, PE-conjugated-anti-Rorγt, BV605-conjugated-anti-IFN-γ, and BV711-conjugated-anti-IL-4 antibodies respectively. After washing, cells were acquired on BD LSR Fortessa (USA). Flow Jo 10 (Tree Star, Woodburn, OR, USA) software was employed to analyze the samples.

### 2.11. Enzyme-Linked Immunosorbent Assay (ELISA) 

The levels of IL-4, IL-6, IL-10, IL-17, IFN-γ, and TNF-α proteins were determined from the collected blood serum samples by using commercially available kits (OptEIA BD, Franklin Lakes, NJ, USA) and adhering to the manufacturer’s instruction and safety precautions. Multimode microplate reader BioTek Synergy H1, BioTek Instruments (USA) was used for readings as per the requirements.

### 2.12. Histologic Analysis

Femoral bones were decalcified for 15 days via the use of 10 % tetrasodium EDTA aqueous solution by placing it on a rocker. Paraffin-embedded sections (5 μm) from each femur were processed for the histological hematoxylin and eosin staining analysis. Sections were imaged using a microscope.

### 2.13. Statistical Analysis of Data 

For performing the statistical analysis of the available data, results were represented as mean ± SEM for at least two independent studies with 6 mice per group. Comparisons were made between three groups, Sham, Ovx, and Ovx + CQ, using one-way analysis of variance (ANOVA), followed by an unpaired student t-test. In comparison to the Ovx group, the statistical significance level of the data was established as *p* ≤ 0.05 (* *p* ≤ 0.05, ** *p* ≤ 0.01, *** *p* ≤ 0.001).

## 3. Results

### 3.1. CQ Inhibits RANKL-Induced Osteoclastogenesis

Firstly, we were interested in investigating the anti-osteoclastogenic potential of CQ under ex vivo conditions. This was achieved by performing RANKL-induced osteoclast development in bone marrow cells (BMCs) using osteoclastogenic medium supplemented with M-CSF and RANKL in the presence or absence of CQ at varied doses, i.e., 10, 20, 50, and 100 μg/mL. The multinucleated mature osteoclasts were identified by TRAP-staining after four days of incubation (Figure 1A). It was observed that CQ treatment greatly reduced the quantity of large multinucleated osteoclasts (Figure 1B). Moreover, the total number of TRAP^+^ osteoclasts were also significantly reduced (*p* ≤ 0.001) in a dose-dependent manner relative to the control group (Figure 1C,D). Additionally, in comparison to the control group, the area of TRAP^+^ multinucleated mature osteoclasts were also markedly diminished in the CQ treatment group (Figure 1E). Further, we employed MTT assay to rule out the hypothesis that the observed decrease in osteoclast differentiation and number is due to CQ-cytotoxicity, and we observed that CQ treatment at different concentrations has no effect on cell viability (Appendix A). These findings clearly suggest that CQ has potent anti-osteoclastogenic properties.

### 3.2. CQ Attenuates the Functional Activity of Osteoclasts

A crucial aspect of mature and functional multinucleated osteoclasts for bone resorption is the formation of F-actin rings. Thus, using an F-actin assay, we next evaluated CQ’s impact on osteoclast’s functional ability. We observed that in comparison to the control group, CQ treatment considerably decreased both the number as well as the area of F-actin rings in a dose-dependent manner (Figure 2A–D). These findings further supported our above results that CQ not only suppressed the formation of osteoclasts but also robustly inhibited the functional activity of osteoclasts. 

### 3.3. CQ Augments Bone Health under Postmenopausal Osteoporotic Conditions

Moving ahead we next evaluated the potential of CQ in reducing the inflammatory bone loss under estrogen-deficient conditions in a preclinical model of osteoporosis, i.e., Ovx. For achieving the same, 8–10 weeks old Balb/c (female mice) were arbitrarily divided into three groups, i.e., Sham, Ovx, and Ovx + CQ group. CQ was given orally to Ovx + CQ group mice for 45 days at a dose of 500 mg/kg/day. At the end of the experiment, all the animals were euthanized, and bone samples were harvested for analysis of various bone parameters (Figure 3A). Firstly, we used SEM to assess CQ’s osteoprotective potential. In comparison to the sham group, SEM image analysis of the femur cortical areas revealed the development of larger Howship’s lacunae/resorption pits, indicating increased osteoclastogenesis in the Ovx group (Figure 3B). Surprisingly, CQ administration in the Ovx group resulted in a large reduction in the Howship’s lacunae and resorption area, thereby indicating reduced osteoclastogenesis (Figure 3B). We used MATLAB (matrix laboratory) analysis to determine the degree of homogeneity in 2D-SEM images to increase our understanding of the relationship between bone loss and bone development. In MATLAB analysis, the blue color denotes a lower correlation or lower bone mass. Contrarily, the red color denotes a higher correlation, or larger bone mass. A thorough examination showed that the Ovx + CQ group had more bone mass than the Ovx group, indicating a higher correlation (Figure 3C). Overall, our SEM data further validates our ex vivo findings that CQ treatment markedly decreased bone resorption in Ovx mice.

### 3.4. CQ Enhances Bone Micro-Architecture and Histomorphometric Indices

To further analyze the potential of CQ in maintaining bone health, we next performed bone micro-architecture and histomorphometric analysis via high-resolution µ-CT, (a gold standard), for calculating various bone morphometric indices. Since, the LV-5 region is crucial in the early detection of osteoporosis or bone loss [15,19], thus we next investigated the effect of CQ in the trabeculae region of LV-5. We observed that the administration of CQ to the Ovx mice group significantly enhanced the bone microarchitecture of the LV-5 trabecular region as compared to the Ovx group (Figure 4A). In addition, the LV-5 trabecular histomorphometric analysis for various parameters such as BV/TV, Tb.Th, Tb.N, Conn.D, Tb.Sp, and Tb.Pf was also considered. Excitingly, it was observed that BV/TV (*p ≤* 0.01), Tb.Th (*p ≤* 0.05), Tb.N (*p ≤* 0.05), and Conn.D (*p ≤* 0.05) parameters were significantly enhanced in the Ovx + CQ group along with a significant decrease in the Tb.Sp (*p ≤* 0.05) and Tb.Pf (*p ≤* 0.05) (Table 1). Next, we monitored the histomorphometric parameters of the trabecular and cortical regions of the femoral and tibial bones, where we observed a significant improvement in the 3D micro-architecture of both the trabecular and cortical regions of the CQ administered Ovx group (Figure 4C, E, G, I). Similar results were also observed in trabecular bone indices, i.e., BV/TV (*p ≤* 0.01), Tb.Th (*p ≤* 0.05), Tb.N (*p ≤* 0.05), and Conn.D (*p ≤* 0.05) parameters, along with a significant decrease in the Tb.Sp (*p ≤* 0.05) and Tb.Pf (*p ≤* 0.05) for femur trabecular regions (Table 1). Likewise, we also observed a significant increase in the histomorphometric parameters of the tibia trabecular region along with an improved 3D bone micro-architecture (Table 1). Notably, administration of CQ significantly improved the bone microarchitecture and histomorphometric indices of the cortical region in both femoral as well as tibial bones relative to the Ovx group (Table 1). These results were further corroborated by decalcified H a E-stained bone sections, wherein femoral sections from the Ovx mice showed a dearth of cancellous and trabecular bone regions and, treatment with CQ significantly reverses the same in Ovx mice (Appendix A). Collectively, our data indicate that CQ treatment dramatically improves both histomorphometric and bone micro-architecture indices and improves bone health in Ovx mice under estrogen-deficient settings.

### 3.5. CQ Improves Bone Mineral Density (BMD) 

Since BMD is regarded as a determinant of osteoporotic fractures, we therefore next assessed the BMDs of both the axial and appendicular sites that bear body weight. µCT permits the measurement of BMDs, and our findings clearly demonstrated significant enhancement in the BMDs of the LV-5, femur, and tibia trabecular as well as the cortical regions of the bones, indicating that administration of CQ significantly improves the mineral content of bones (Figure 4B,D,F,H,J). Altogether, both our µ-CT analysis and BMD data clearly suggest that CQ administration improves BMDs and thus enhances bone health in Ovx mice.

### 3.6. CQ Augments Bone Health Via Modulating Immunoporotic Cells 

Recent studies from our group have demonstrated the crucial role of immune systemin the pathophysiology of osteoporosis, which is now referred to as “Immunoporosis”. Numerous studies along with our own had established the anti-osteoclastogenic role of Th1, Th2, Tregs, and Bregs along with the osteoclastogenic role of Th17 in osteoporosis. Consequently, we next investigated the immuno-modulatory capability of CQ in enhancing bone health. We observed that CQ administration drastically increased the population of anti-osteoclastogenic Th1 (CD4^+^IFN-γ^+^) (Figure 5A–H), Th2 (CD4^+^IL-4^+^) (Figure 6A–H), Tregs (CD4^+^Foxp3^+^) (Figure 7A–H), and Bregs (CD19^+^CD1d^hi^CD5^+^) (Figure 8A–H; Appendix A) in various lymphoid organs such as BM (prime site of osteoclastogenesis), SP, MLN, and PP. We further evaluated the percentage of osteoclastogenic Th17 cells (CD4^+^ Rorγt^+^) via flow cytometry in all the groups. Intriguingly, we found that oral treatment of CQ significantly reduced the amount of inflammatory Th17 cells in several lymphoid organs, including the BM, SP, MLN, and PP (Figure 9A–H). Altogether, our data convincingly demonstrate that oral administration of CQ improves bone health via regulating immunoporotic cells.

### 3.7. CQ Skews the Cytokine Balance under Estrogen-Deficient Conditions 

Osteoclastogenic cytokines secreted by Th17 cells (IL-6, IL-17, TNF-α) and anti-osteoclastogenic cytokines secreted by Th1, Th2, Tregs, and Bregs, respectively, (IFN-γ, IL-4, and IL-10) are the major players in bone remodeling under estrogen-deficient inflammatory conditions (15, 19). Thus, we next analyzed the levels of these cytokines in the sera of mice in all the groups. When compared to the Ovx mice group, we discovered that administering CQ significantly decreased the levels of osteoclastogenic cytokines like IL-6, IL-17, and TNF-α along with significantly (*p* < 0.05) increasing the levels of anti-osteoclastogenic cytokines like IL-4, IL-10, and IFN-γ. (Figure 10). Taken together our data establishes that *Cissus quadrangularis* administration skews the cytokine balance even under estrogen-deficient conditions and thus enhances bone health. 

## 4. Discussion

Over the past few years, plant-based formulations are gaining much importance with evolving pieces of evidence indicating the therapeutic role of phytoconstituents in maintaining bone-related diseases [23,38], especially via regulating the host osteoimmune system [39]. In the present investigation, we sought to elucidate the osteoprotective effect of CQ on bone health under both ex vivo and in vivo conditions. We discovered that CQ treatment significantly reduced the differentiation and functional activity of osteoclasts. Moving forward, we next investigated the osteoprotective potential of CQ under in vivo conditions. We discovered that oral treatment of CQ significantly improves bone mass by enhancing the microarchitecture of the LV-5, femoral, and tibial bones’ trabecular and cortical sections (SEM and μCT data). These data are in line with the findings by Banu et al. and Ramachandran et al. which demonstrated the osteoprotective potential of CQ in the cancellous and cortical regions of femoral and proximal tibial bones in mice and Wistar rats (Banu 2012; Ramachandran 2021). In addition, a study highlighted the potential of CQ in decreasing both pain and swelling along with accelerating the healing of fractured jaws (Hemal 2015). Moreover, our histology data clearly revealed that CQ administration has no cytotoxic effect on different organs of the body, and also no change in body weight was observed after CQ administration (Appendix A).

BMD is a well-established determining factor for assessing future incidences of fragility fractures in osteoporosis [40,41]. Various earlier investigations with small animal models have highlighted the bone-healing and osteoprotective properties of CQ [42,43]. The current study’s findings provide additional evidence that CQ treatment enhances BMDs in both trabecular and cortical regions of the LV-5, femoral, and tibial bones. Importantly, to date no study has ever explored and correlated the osteoprotective and immunoporotic mechanisms of CQ in bone health. Therefore, the present study for the first time delineated CQ’s capacity to improve bone health via modulating the host osteoimmune system.

Our group along with others has already established the pioneering role of Tregs-Th17 cells under osteoporotic conditions. Recently, we also discovered the role of Bregs in osteoporosis which further modulates the pivotal balance of Tregs and Th17 cells. These studies thus highlight and establish the strategic role of “Breg-Treg-Th17” cell axis in the therapeutic management of osteoporosis. In support of this, a recent study from our group has already reported that probiotic *Bifidobacterium longum* ameliorates bone loss by modulating the “Breg-Treg-Th17” axis [22]. Moving ahead, we thus explored the potential of CQ in modulating the immunoporotic “Breg-Treg-Th17” cell axis in enhancing bone health. Our flow cytometry findings demonstrated that oral treatment of CQ improve bone health even under oestrogen deficient conditions by significantly increasing anti-osteoclastogenic Bregs and Tregs along with reducing the proportion of osteoclastogenic Th17 cells. It is well known that the cytokines produced by immune cells, including the anti-osteoclastogenic cytokines IFN-γ, IL-4, and IL-10 and the osteoclastogenic cytokines TNF-α, IL-6, and IL-17, influence bone remodeling [15,19]. Importantly, the immunoporotic potential of CQ was further confirmed by serum cytokine analysis, which revealed that oral administration of CQ significantly decreased the osteoclastogenic cytokines viz. IL-6, IL-17 (Th17 cells), and TNF-α, and increased the anti-osteoclastogenic cytokine, i.e., IL-10 (Bregs and Tregs), IL-4 (Th2), and IFN-γ (Th1). Collectively, the present study for the first time robustly establishes the osteo-protective and bone healing properties of CQ via modulating the “Immunoporotic” cells even under estrogen deficient inflammatory bone-loss conditions in preclinical mice models of osteoporosis (Figure 11).

## 5. Conclusions

The current study clearly establishes the osteoprotective role of CQ in osteoporosis via therapeutically targeting and modulating the crucial osteoprotective “Breg-Treg-Th17” cell axis. Thus, the results of our study pave the way for future clinical employability of CQ in the management and treatment of a variety of inflammatory bone loss conditions including osteoporosis. However, more studies are still warranted to further validate the future perspective of CQ administration as an anabolic drug in the prevention and management of osteoporosis.

## Figures and Tables

**Figure 1 cells-12-00216-f001:**
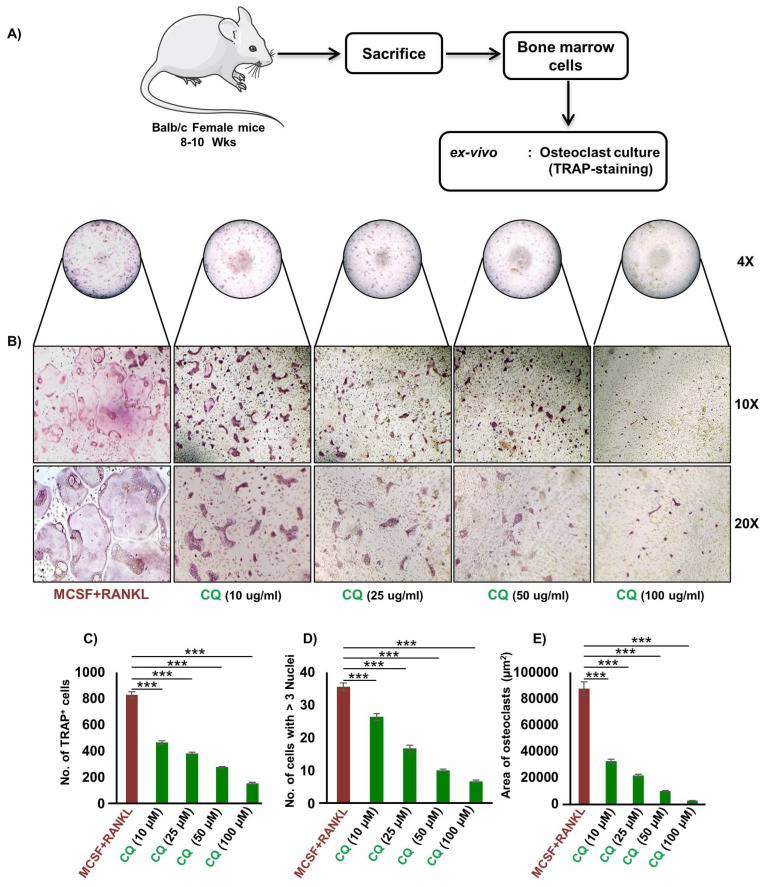
**CQ inhibits osteoclastogenesis in a dose-dependent manner:** (**A**) Experimental layout followed for ex vivo osteoclast culture from mice bone marrow cells (BMCs). Large multinucleated cells TRAP positive osteoclasts with ≥ 3 nuclei were considered matured osteoclasts. (**B**) Photomicrographs at 4×, 10×, and 20× indicate TRAP staining in osteoclasts. (**C**) Number of TRAP-positive cells. (**D**) Number of TRAP-positive cells with ≥ 3 nuclei. (**E**) Area of osteoclasts. ANOVA was used to analyse the results, and then unpaired Student t-tests were used to compare the indicated groups. Values are expressed as mean ± SEM, and equivalent outcomes were observed in two independent experiments. For the indicated groups, *p* ≤ 0.05 *** *p* ≤ 0.001) was used as the cutoff for statistical significance. In the bar graphs, red is indicating the positive control and green is indicating the treatment groups.

**Figure 2 cells-12-00216-f002:**
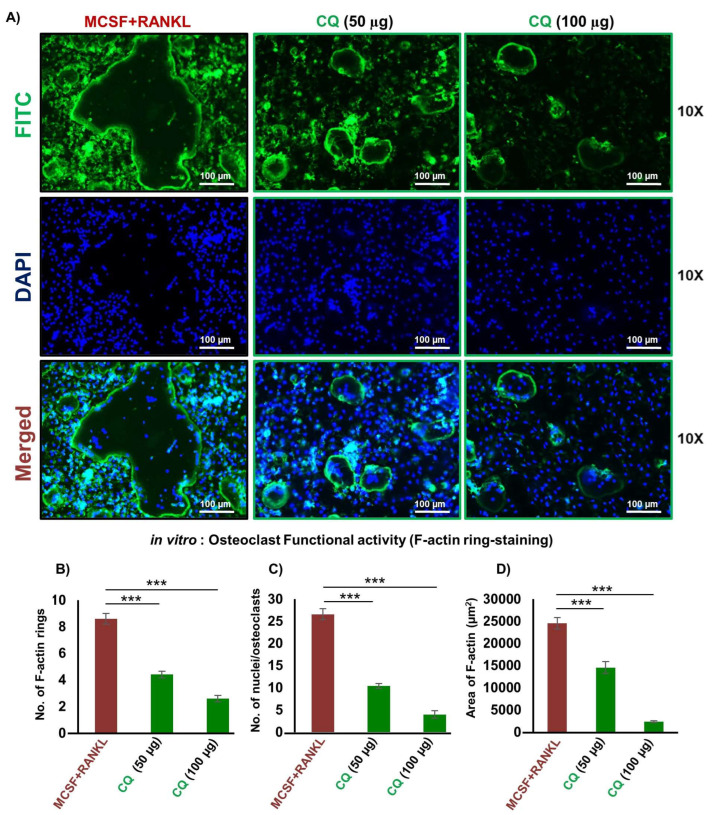
**CQ inhibits the functional activity of osteoclasts:** (**A**) FITC-conjugated phalloidin and DAPI were used to stain F-Actin and the nuclei, respectively. At a 10X magnification, images were taken using a fluorescent microscope (Imager.Z2 Zeiss microscope). (**B**) Number of F-actin rings. (**C**) Number of nuclei per osteoclast. (**D**) Area of F-actin rings. ANOVA was used to analyse the results, and then unpaired Student t-tests were used to compare the indicated groups. Values are expressed as mean ± SEM, and equivalent outcomes were observed in two independent experiments. For the indicated groups, *p* 0.05 *** *p* ≤ 0.001) was used as the cutoff for statistical significance. In the bar graphs, red is indicating the positive control and green is indicating the treatment groups.

**Figure 3 cells-12-00216-f003:**
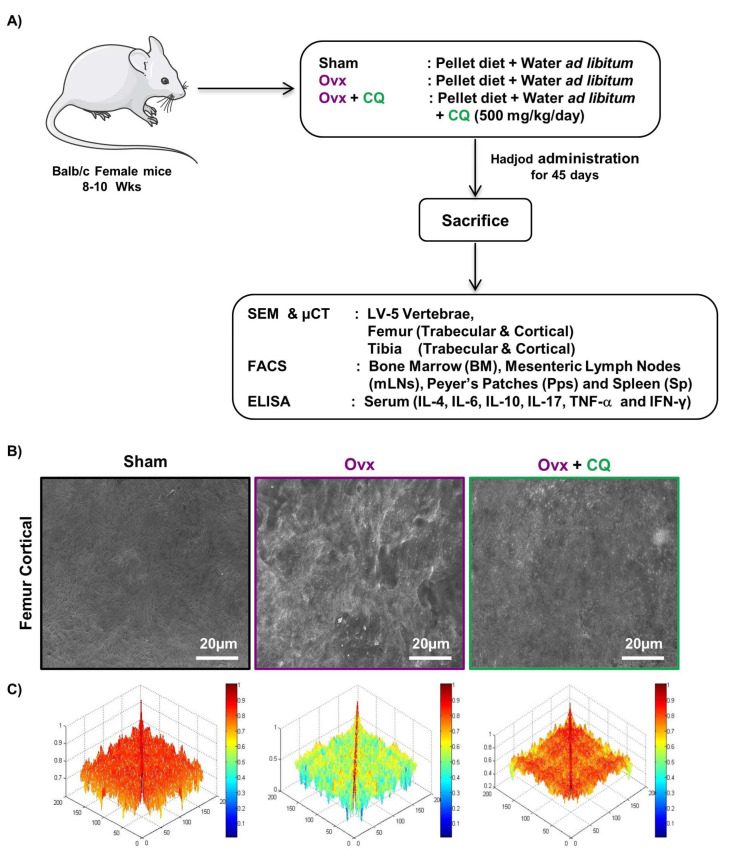
**CQ administration attenuates bone loss in Ovx mice:** (**A**) Experimental layout followed for in vivo studies. (**B**) Two dimensional SEM images of the femur cortical region. (**C**) Two dimensional MATLAB analysis of SEM images. ANOVA was used to analyse the results, and then unpaired Student t-test were used to compare the indicated groups. Values are expressed as mean ± SEM (n = 6) in the above experiment and similar results were obtained in two independent experiments.

**Figure 4 cells-12-00216-f004:**
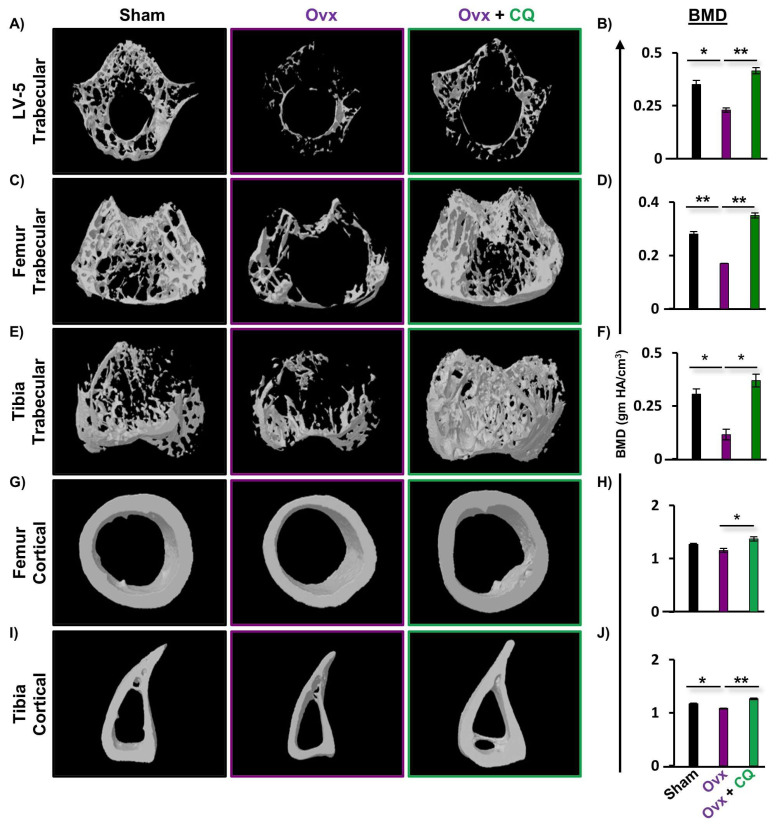
**CQ administration improves trabecular bone microarchitecture:** Three dimensional µ-CT reconstruction of LV-5 (trabecular), femur (trabecular and cortical), and tibia (trabecular and cortical) and graphical presentation of BMD (gm HA/cm^3^) of all the following groups. (**A**) Bone microarchitecture of LV-5 trabecular region. (**B**) BMD of LV-5 trabecular. (**C**) Bone microarchitecture of femur trabecular region. (**D**) BMD of femur trabecular region. (**E**) Bone microarchitecture of the tibia trabecular region. (**F**) BMD of the tibia trabecular region. (**G**) Bone microarchitecture of the femur cortical region. (**H**) BMD of the femur cortical region. (**I**) Bone microarchitecture of the tibia cortical region. (**J**) BMD of the tibia cortical region. BV/TV; bone volume/tissue volume ratio, Tb.Th; trabecular thickness, Tb.Sp; trabecular separation, Tt.Ar; total cross-sectional area inside the periosteal envelope, Ps.Pm; periosteal perimeter, Ct.Th; average cortical thickness and BMD; bone mineral density. ANOVA was used to analyse the results, and then unpaired Student t-tests were used to compare the indicated groups. Values are expressed as mean ± SEM, and equivalent outcomes were observed in two independent experiments. For the indicated groups, *p* 0.05 (* *p* ≤ 0.05, ** *p* ≤ 0.01) was used as the cutoff for statistical significance. In the figure, black color is indicating control group, purple color is indicating osteoporotic group and green color is highlighting osteoporotic group treated with CQ.

**Figure 5 cells-12-00216-f005:**
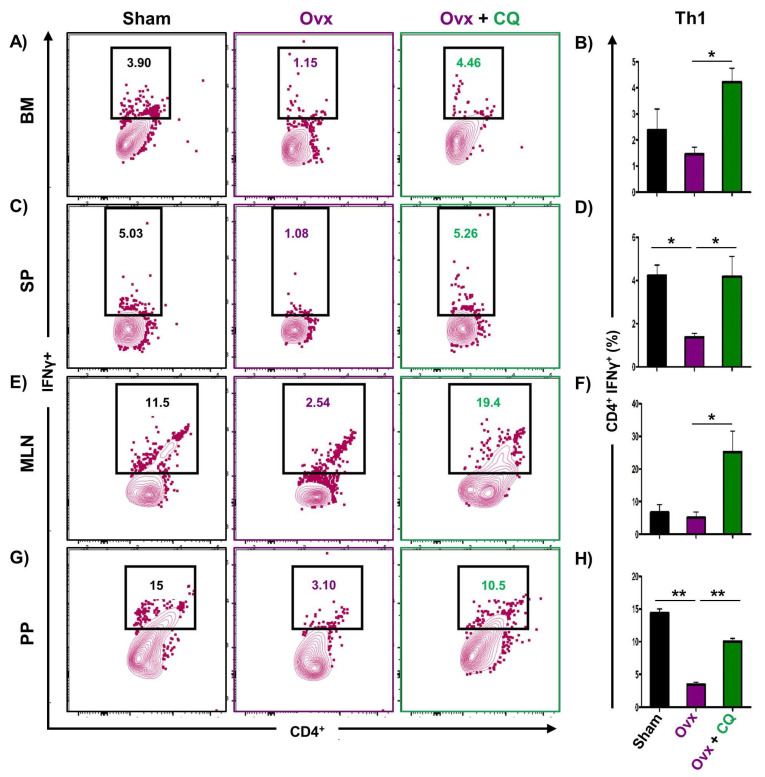
**CQ administration enhances Th1 cells in Ovx mice:** Immune cells harvested from the lymphoid organs of all the groups were analyzed by flow cytometry for the percentage of CD4^+^IFNγ^+^ Th1 immune cells. (**A**) Contour plot representing the percentage of CD4^+^IFNγ^+^ Th1 in BM. (**B**) Bar graphs representing the percentage of CD4^+^IFNγ^+^ Th1 in BM. (**C**) Contour plot representing the percentage of CD4^+^IFNγ^+^ Th1 in SP. (**D**) Bar graphs representing the percentage of CD4^+^IFNγ^+^ Th1 in SP. (**E**) Contour plot representing the percentage of CD4^+^IFNγ^+^ Th1 in MLN. (**F**) Bar graphs representing the percentage of CD4^+^IFNγ^+^ Th1 in MLN. (**G**) Contour plot representing the percentage of CD4^+^IFNγ^+^ Th1 in PP. (**H**) Bar graphs representing the percentage of CD4^+^IFNγ^+^ Th1 in PP. ANOVA was used to analyse the results, and then unpaired Student t-tests were used to compare the indicated groups. Values are expressed as mean ± SEM, and equivalent outcomes were observed in two independent experiments. For the indicated groups, *p* 0.05 (* *p* ≤ 0.05, ** *p* ≤ 0.01) was used as the cutoff for statistical significance.

**Figure 6 cells-12-00216-f006:**
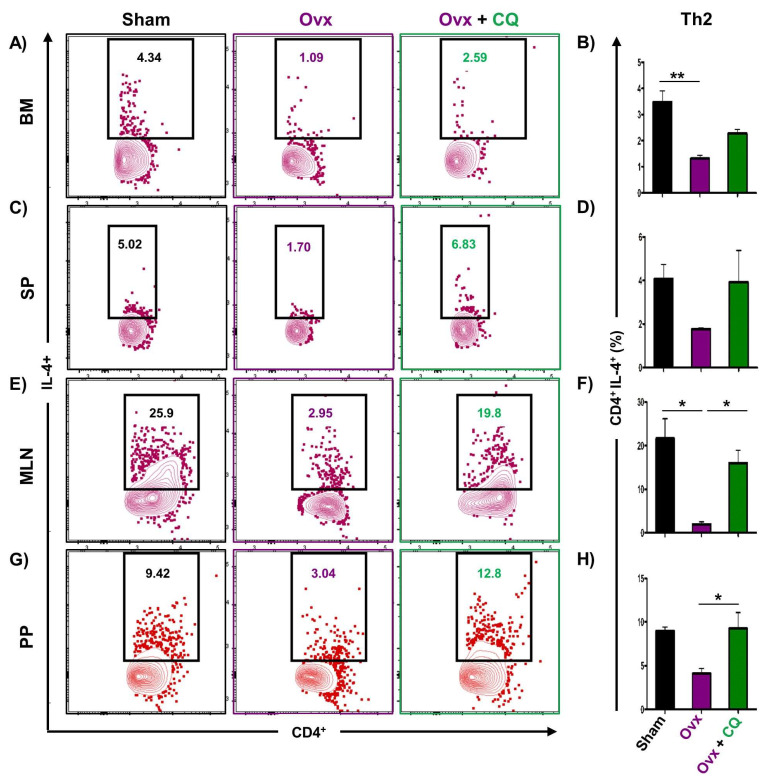
**CQ administration enhances Th2 cells in Ovx mice:** Immune cells harvested from the lymphoid organs of all the groups were analyzed by flow cytometry for the percentage of CD4^+^IL-4^+^ Th2 immune cells. (**A**) Contour plot representing the percentage of CD4^+^IL-4^+^ Th2 in BM. (**B**) Bar graphs representing the percentage of CD4^+^IL-4^+^ Th2 in BM. (**C**) Contour plot representing the percentage of CD4^+^IL-4^+^ Th2 in SP. (**D**) Bar graphs representing the percentage of CD4^+^IL-4^+^ Th2 in SP. (**E**) Contour plot representing the percentage of CD4^+^IL-4^+^ Th2 in MLN. (**F**) Bar graphs representing the percentage of CD4^+^IL-4^+^ Th2 in MLN. (**G**) Contour plot representing the percentage of CD4^+^IL-4^+^ Th2 in PP. (**H**) Bar graphs representing the percentage of CD4^+^IL-4^+^ Th2 in PP. ANOVA was used to analyse the results, and then unpaired Student t-tests were used to compare the indicated groups. Values are expressed as mean ± SEM, and equivalent outcomes were observed in two independent experiments. For the indicated groups, *p* 0.05 (* *p* ≤ 0.05, ** *p* ≤ 0.01) was used as the cutoff for statistical significance.

**Figure 7 cells-12-00216-f007:**
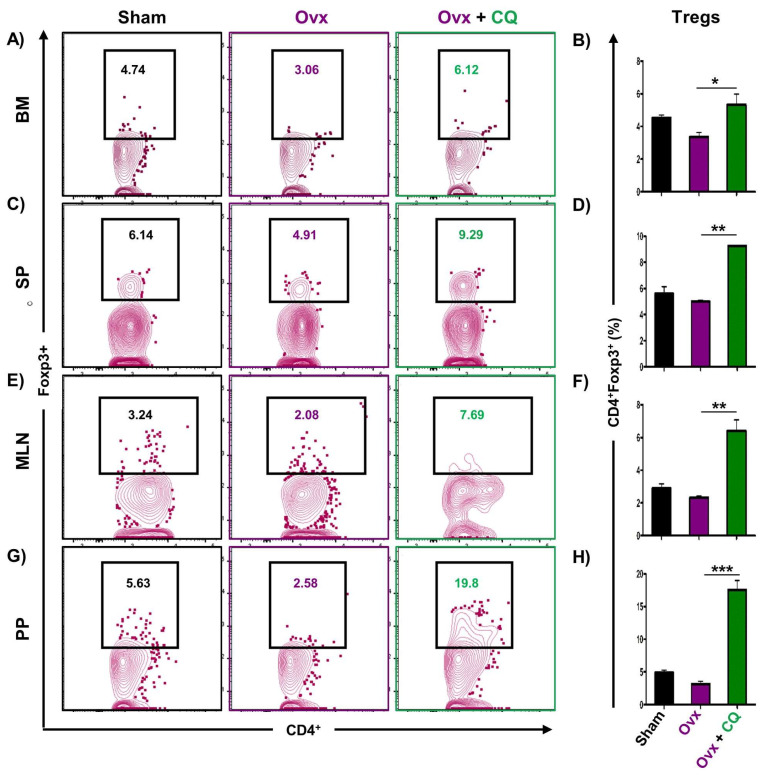
**CQ administration enhances Tregs cells in Ovx mice:** Immune cells harvested from the lymphoid organs of all the groups were analyzed by flow cytometry for the percentage of CD4^+^Foxp3^+^ Treg immune cells. (**A**) Contour plot representing the percentage of CD4^+^Foxp3^+^ Treg in BM. (**B**) Bar graphs representing the percentage of CD4^+^Foxp3^+^ Treg in BM. (**C**) Contour plot representing the percentage of CD4^+^Foxp3^+^ Treg in SP. (**D**) Bar graphs representing the percentage of CD4^+^Foxp3^+^ Treg in SP. (**E**) Contour plot representing the percentage of CD4^+^Foxp3^+^ Treg in MLN. (**F**) Bar graphs representing the percentage of CD4^+^Foxp3^+^ Treg in MLN. (**G**) Contour plot representing the percentage of CD4^+^Foxp3^+^ Treg in PP. (**H**) Bar graphs representing the percentage of CD4^+^Foxp3^+^ Treg in PP. ANOVA was used to analyse the results, and then unpaired Student *t*-tests were used to compare the indicated groups. Values are expressed as mean ± SEM, and equivalent outcomes were observed in two independent experiments. For the indicated groups, *p* 0.05 (* *p* ≤ 0.05, ** *p* ≤ 0.01, *** *p* ≤ 0.001) was used as the cutoff for statistical significance.

**Figure 8 cells-12-00216-f008:**
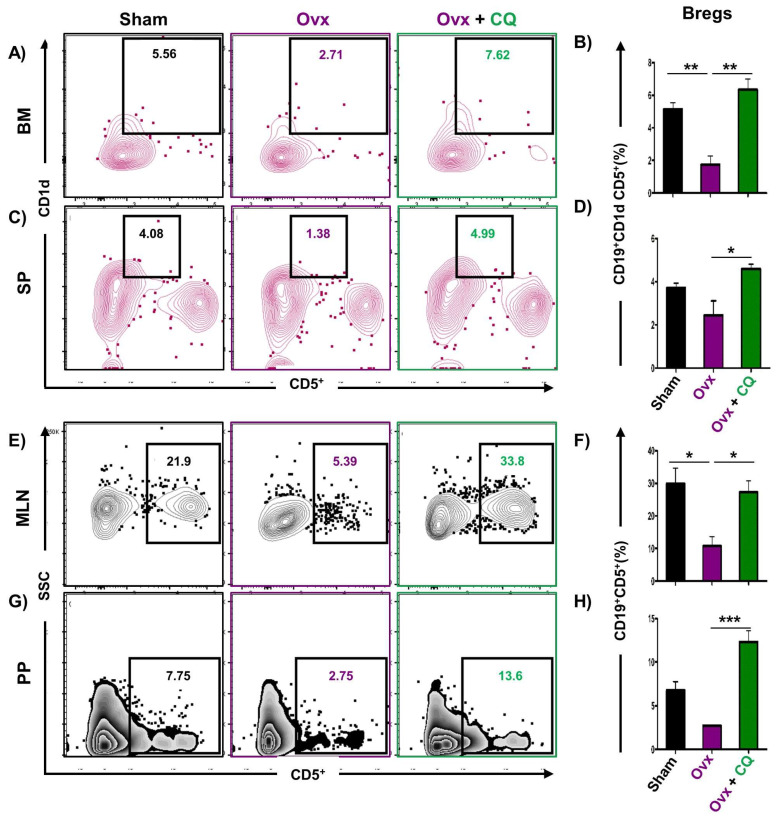
**CQ intake enhances Bregs in Ovx mice:** Cells harvested from the lymphoid organs of all the groups were analyzed by flow cytometry for the percentage of CD19^+^CD1d^hi^CD5^+^ Bregs immune cells. (**A**) Contour plot representing the percentage of CD19^+^CD1d^hi^CD5^+^ Bregs in BM. (**B**) Bar graphs representing the percentage of CD19^+^CD1d^hi^CD5^+^ Bregs in BM. (**C**) Contour plot representing the percentage of CD19^+^CD1d^hi^CD5^+^ Bregs in SP. (**D**) Bar graphs representing the percentage of CD19^+^CD1d^hi^CD5^+^ Bregs in SP. (**E**) Contour plot representing the percentage of CD19^+^CD5^+^ Bregs in MLN. (**F)** Bar graphs representing the percentage of CD19^+^CD5^+^ Bregs in MLN. (**G**) Contour plot representing the percentage of CD19^+^CD5^+^ Bregs in PP. (**H**) Bar graphs representing the percentage of CD19^+^CD5^+^ Bregs in PP. ANOVA was used to analyse the results, and then unpaired Student t-tests were used to compare the indicated groups. Values are expressed as mean ± SEM, and equivalent outcomes were observed in two independent experiments. For the indicated groups, *p* 0.05 (* *p* ≤ 0.05, ** *p* ≤ 0.01, *** *p* ≤ 0.001) was used as the cutoff for statistical significance.

**Figure 9 cells-12-00216-f009:**
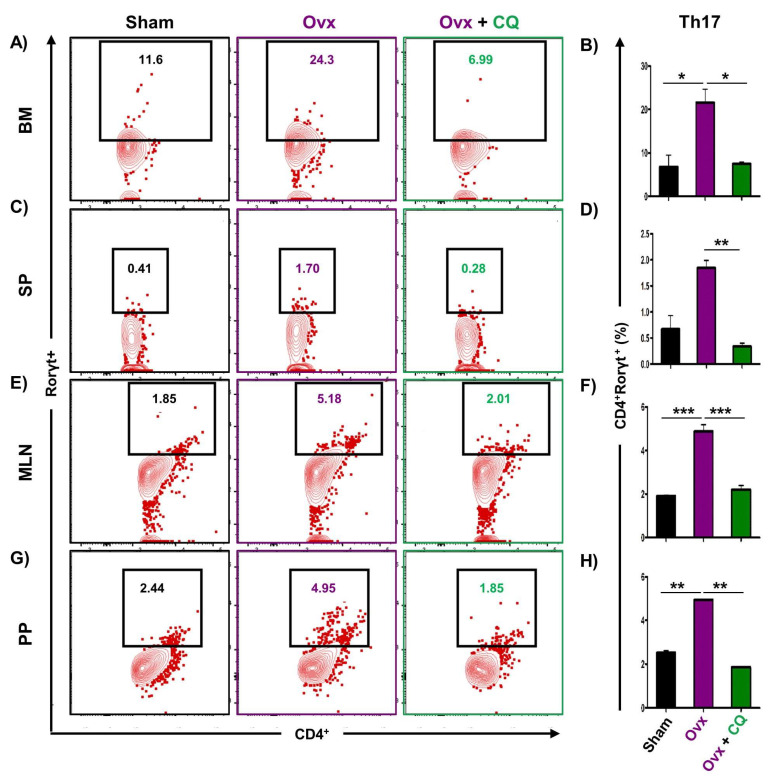
**CQ administration decreases the Th17 cell population in Ovx mice:** Cells harvested from the lymphoid organs of all the groups were analyzed by flow cytometry for the percentage of CD4^+^Rorγt^+^ Th17 immune cells. (**A**) Contour plot representing the percentage of CD4^+^Rorγt^+^ Th17 in BM. (**B**) Bar graphs representing the percentage of CD4^+^Rorγt^+^ Th17 in BM. (**C**) Contour plot representing the percentage of CD4^+^Rorγt^+^ Th17 in SP. (**D**) Bar graphs representing the percentage of CD4^+^Rorγt^+^ Th17 in SP. (**E**) Contour plot representing the percentage of CD4^+^Rorγt^+^ Th17 in MLN. (**F**) Bar graphs representing the percentage of CD4^+^Rorγt^+^ Th17 in MLN. (**G**) Contour plot representing the percentage of CD4^+^Rorγt^+^ Th17 in PP. (**H**) Bar graphs representing the percentage of CD4^+^Rorγt^+^ Th17 in PP. ANOVA was used to analyse the results, and then unpaired Student t-tests were used to compare the indicated groups. Values are expressed as mean ± SEM, and equivalent outcomes were observed in two independent experiments. For the indicated groups, *p* 0.05 (* *p* ≤ 0.05, ** *p* ≤ 0.01, *** *p* ≤ 0.001) was used as the cutoff for statistical significance.

**Figure 10 cells-12-00216-f010:**
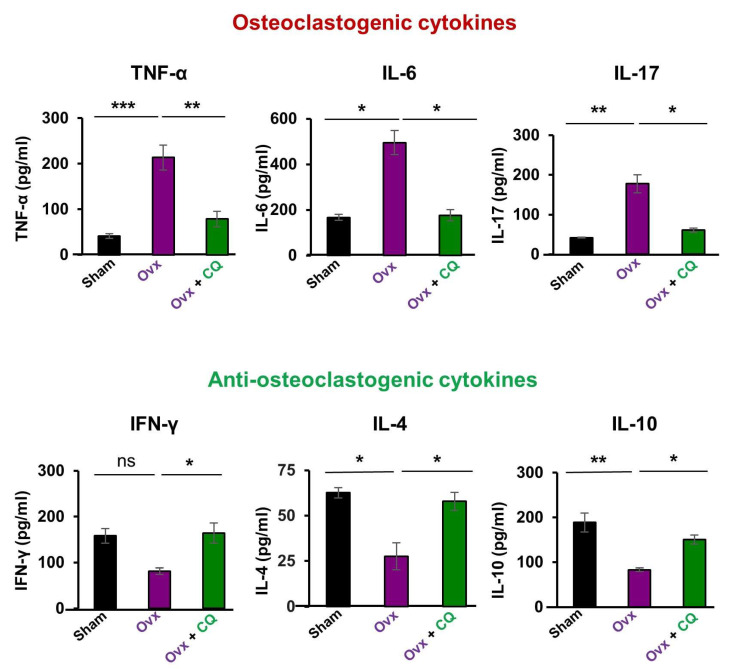
**CQ skews cytokine balance in Ovx mice:** Serum samples from all the three groups viz., Sham, Ovx, and Ovx + CQ were analyzed for levels of pro and anti-inflammatory cytokines by ELISA. ANOVA was used to analyse the results, and then unpaired Student t-tests were used to compare the indicated groups. Values are expressed as mean ± SEM, and equivalent outcomes were observed in two independent experiments. For the indicated groups, *p* 0.05 (* *p* ≤ 0.05, ** *p* ≤ 0.01, *** *p* ≤ 0.001) was used as the cutoff for statistical significance.

**Figure 11 cells-12-00216-f011:**
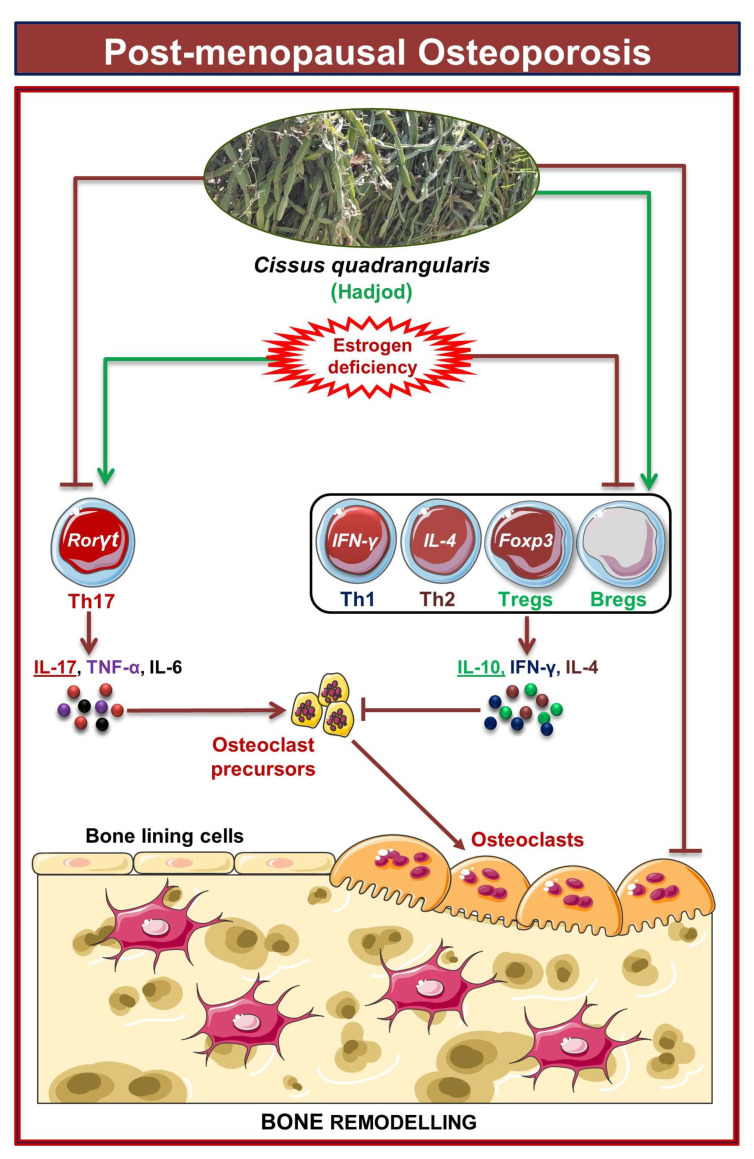
**Results summary:** In PMO conditions, oral administration of CQ improves bone health by reducing the percentage of Th17 cells along with simultaneously increasing the percentages of Th1, Th2, Tregs, and Bregs immune cells in various lymphoid organs of the Ovx mice. The image was produced using Servier Medical Art, a resource made available by Servier and distributed under a Creative Commons Attribution 3.0 Unported license (https://smart.servier.com).

**Table 1 cells-12-00216-t001:** Bone histomorphometric indices.

Bone Parameters	Sham	Ovx	Ovx + CQ
LV-5			
BV/TV (%)	68.8 ± 0.1	56.4 ± 2.5	83.35 ± 1.35 **
Tb. Th (mm)	0.05 ± 0	0.045 ± 0.005	0.055 ± 0.005 *
Tb. Sp (mm^−3^)	0.038 ± 0.002	0.058 ± 0.002	0.037 ± 0.0015 *
Conn. D (mm^−3^)	198.38 ± 40.22	93.87 ± 27.52	223.153 ± 17.29 *
Tb. N (mm^−1^)	9.8 ± 0.1	13.45 ± 0.45	16.95 ± 1.75 *
Tb. Pf (%)	15.05 ± 2.65	21.25 ± 0.25	7.05 ± 0.65 *
Femur Trabecular			
BV/TV (%)	85.3 ± 2.6	63.35 ± 1.25	89.15 ± 0.75 **
Tb. Th (mm)	0.05 ± 0	0.041 ± 0.0015	0.07 ± 0 ***
Tb. Sp (mm^−3^)	0.2 ± 0	0.4 ± 0	0.75 ± 0.0015 **
Conn. D (mm^−3^)	185.51 ± 14.85	83.39 ± 7.75	179.04 ± 9.13 ***
Tb. N (mm^−1^)	7.36 ± 4.78	3.56 ± 2.21	11.4 ± 7.20 *
Tb. Pf (%)	14.4 ± 0.62	19.03 ± 1.26	12.53 ± 1.65 **
Tibia Trabecular			
BV/TV (%)	81.65 ± 2.75	62.15 ± 2.05	86.35 ± 2.85 *
Tb. Th (mm)	0.07 ± 0	0.046 ± 0.005	0.00071 ± 0 ***
Tb. Sp (mm^−3^)	0.037 ± 0.0035	0.046 ± 0.0015	0.025 ± 0.005 **
Conn. D (mm^−3^)	50.33 ±	27.89 ±	86.30 ± 53.60 *
Tb. N (mm^−1^)	9.13 ± 1.73	6.7 ± 0.22	13.5 ± 1.20 **
Tb. Pf (%)	13.63 ± 2.35	20 ± 1.79	12.86 ± 0.96 *
Femur Cortical			
Tt. Ar (mm^2^)	1.35 ± 0.01	0.8 ± 0.1	2.03 ± 0.06 **
T. Pm (mm)	4.38 ± 0.025	3.22 ± 0.045	5.4 ±0.08 ***
Ct. Th (mm)	0.16 ± 0.005	0.13 ± 0.005	0.17 ± 0.005 *
Ct. Ar (mm^2^)	0.59 ± 0.57	0.54 ± 0.55	0.7 ± 0.08 *
B. Pm (mcm)	7.60 ± 0.03	6.57 ± 0.56	9.39 ± 0.77
*J* (mm^4^)	0.21 ± 0.02	0.13 ± 0.04	0.26 ± 0.08
Tibia Cortical			
Tt. Ar (mm^2^)	0.975 ± 0.015	0.65 ± 0.05	1.39 ± 0.05 **
T. Pm (mm)	4.51 ± 0	4.68 ± 0.02	5.57 ± 0.115 **
Ct. Th (mm)	0.14 ± 0.01	0.11 ± 0.01	0.16 ± 0.01 *
Ct. Ar (mm^2^)	0.53 ± 0.05	0.46 ± 0.02	0.66 ± 0.02 **
B. Pm (mcm)	5.27 ± 3.22	4.72 ± 2.90	6.77 ± 4.15 *
*J* (mm^4^)	0.14 ± 0.02	0.13 ± 0.01	0.31 ± 0.03 *

Table 1 Bone histomorphometric indices Trabecular (LV-5, femur, and tibia) and cortical (femur and tibia) bones were studied using bone histomorphometry. Histomorphometric measurements of the LV-5, femur, and tibia in the Sham, Ovx, and Ovx + CQ groups. BV/TV, none volume/tissue volume ratio; Tb.Th, trabecular thickness; Tb.Sp, trabecular separation; Conn.D, connectivity density; Tb.N, trabecular number; Tb.Pf, trabecular pattern factor; Tt.Ar, total cross-sectional area; T.Pm, total cross-sectional perimeter; Ct.Ar, cortical bone area; B.Pm, bone perimeter; Ct.Th, average cortical thickness and *J,* polar moment of inertia (MMI). ANOVA was used to analyse the results, and then unpaired Student t-tests were used to compare the indicated groups. Values are expressed as mean ± SEM, and equivalent outcomes were observed in two independent experiments. For the indicated groups, *p* 0.05 (* *p* ≤ 0.05, ** *p* ≤ 0.01, *** *p* ≤ 0.001) was used as the cutoff for statistical significance.

## Data Availability

The data presented in this study are available on request from the corresponding author.

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
