# Peer review of "Cissus quadrangularis (Hadjod) Inhibits RANKL-Induced Osteoclastogenesis and Augments Bone Health in an Estrogen-Deficient Preclinical Model of Osteoporosis Via Modulating the Host Osteoimmune System"

_cells, 2023, doi:10.3390/cells12020216_

Round 1

Reviewer 1 Report

Indeed, osteoporosis is a chronic inflammatory bone loss condition, which the skeletal microarchitecture is impaired. This paper is good for readers of this journal. However, I have several concerns prior the acceptance for publication:

-Abstract: To add "numbers" in the data section.

-Introduction: What is hypothesis and aim? It is not clear.

-Methods: What is statistical package used to do analyses? 

-Results: To add the charcterization of rodents, such as body mass and muscle mass and strenght.

-Discussion: Please, to start the discussion including the main findings of manuscript and so the discussion with literature.

-Figure 11 is very GREAT! Congratulations!!!!

Author Response

Kindly refer to the attached file.

Reviewer 2 Report

Overall comments for the authors:

I commend you on your submission, and thank you for your efforts.    This is a fantastic contribution, the methodologies are complete, and it contributes substantially to the literature revealing some fairly profound effects of this traditional Ayurvedic compound. Overall, I feel that the manuscript is acceptable for publication after editing as I have made comments below:

Line by line issues and manuscript critique:

TITLE: Does the title clearly portray the subject and purpose of the study?

Title is fine.

    ABSTRACT: Doers the abstract accurately reflect the study? Are all pertinent finds included?

Line 21 -- my understanding of osteoporosis is actually that it is related more to bone quality and not quantity, so I would not describe osteoporosis as a ‘bone loss” condition personally.

Line 28 – take out “interestingly”, opinion word

Line 33- Immunoporotic – check word use here, would this not infer that CQ would induce osteoporotic conditions? In reality it confers the opposite

    INTRODUCTION: What is the authors' original research question, and does their study support or fulfill it?

The author's research question is whether or not CQ is successful in decrease in osteoporotic conditions seen in a postmenopausal mouse model. Their study does indeed support their question.

Line 47 – “which frequently resulting” -- needs English language editing.

Line 53 – phrased oddly, would rewrite

Line 54 - add an “and” before the word deficiency so that this sentence makes sense.

Line 59 - start this sentence off differently, it reads clunky as is.

Line 62 -- full sentence needs to be rewritten.

***As I am not and English language editor, I will end my English language edits with the above, and will no longer comment on any changes, but this manuscript requires extensive English language editing. It is generally very well written, but just needs to be combed over one more time to make sure that everything reads appropriately to an English language reader.***

Line 72 - perhaps the word counterintuitively is better here than surprisingly.

Overall, the introduction is really good aside from the English language at its that I mentioned above. It does a good job of outlining the background literature in the field, and describing CQ. You all have done a great job of building on your previous work.

    METHODS: Was the research method or study design appropriate? Is it presented sufficiently so that other researchers can duplicate them? Are the sample sizes adequate? Are the statistical analyses appropriate and correct?

Methods are well described and complete. other researchers would be able to duplicate them.

Line 117 – how were stems determined to be of excellent quality?

I do not see a power analysis a priori for sample size determinations. You were able to determine statistically significant findings, so obviously you were appropriately powered, but still this is a deficiency of the study if no power analysis was done.

    RESULTS: Do the results answer the original research questions, as demonstrated in the Results section and tables and figures?

Line 253 - important and clever realization to ensure that the high dose CQ is not just causing cytotoxicity

Line 267 -- I would remove all these uses of the word interestingly and just present the data as is. Those results should not have your opinion statements, the discussion is where you can discuss your own opinion.

Line 365 – again no interestingly

    DISCUSSION: Is the Discussion balanced? Does it put the results in context? Do the authors acknowledge the limitations of the study?

Line 456 – “the” majority

Line 457 - this statement needs a citation and needs to be qualified, this is a strong statement to make about the majority of osteoporosis medications…

Line 458 - do you mean to say along with less side effects?

Overall, the discussion is OK but it is a little bit short and could be beefed up substantially. What other work has already evaluated CQ?

    CONCLUSIONS: Are the conclusions supported by the study findings? Does the study provide new, unique, or confirmatory findings? Will the findings be of interest to clinicians or to the public?

Conclusions are fine and are well founded by the study.

    TABLES AND FIGURES: Are all data presented in the text and tables and figures consistent? Do the tables clearly present information not easily summarized in the text of the paper? Are all of the tables necessary? Are the figures necessary and appropriate? Are they of high quality and clearly labeled? Can any be deleted?

Tables and figures are impressive and are well labeled, I did not find any inconsistencies with labeling etc.

    REFERENCES: Is the References section complete, or is it excessive? Does it include all of the necessary current, relevant sources

Fine

Author Response

Kindly refer to the attached file.
